# Indigenous, Yellow-Feathered Chickens Body Measurements, Carcass Traits, and Meat Quality Depending on Marketable Age

**DOI:** 10.3390/ani12182422

**Published:** 2022-09-14

**Authors:** Chunyou Yuan, Yong Jiang, Zhixiu Wang, Guohong Chen, Hao Bai, Guobin Chang

**Affiliations:** 1Key Laboratory for Animal Genetics & Molecular Breeding of Jiangsu Province, College of Animal Science and Technology, Yangzhou University, Yangzhou 225009, China; 2Joint International Research Laboratory of Agriculture and Agri-Product Safety, The Ministry of Education of China, Institutes of Agricultural Science and Technology Development, Yangzhou University, Yangzhou 225009, China

**Keywords:** yellow-feathered chicken, body measurement, slaughter performance, carcass appearance, meat quality, marketable age

## Abstract

**Simple Summary:**

With an increasing trend in chilled chicken sale and consumers’ preference for indigenous chickens in China, commercial, yellow-feathered chickens are confronted with some carcass trait challenges after slaughter, and age is considered to be an important factor in the quality of chickens. Therefore, we compared two marketable ages (90 days and 100 days) on body measurements, slaughter performance, carcass traits, and meat quality in indigenous, yellow-feathered chickens. This current work shows that extending the marketing age to 100 days improves slaughter performance and meat quality in terms of protein and intramuscular fat contents, but shortening the marketing age to 90 days enhances the quality of carcass characteristics such as follicle density and skin color, especially the spotted skin proportions, and improves meat quality traits such as pH and shear force. The results indicated that chickens on day 90 have a better carcass appearance, which is one of the most intuitive references and will alleviate carcass trait problems in the chilled sale of chickens. We concluded that adjusting the marketable age can be an effective method to drive chilled sales and helps producers to satisfy consumers from different regions in indigenous, yellow-feathered chickens.

**Abstract:**

Given an increasing trend in slaughter and chilling for the sale of chickens in China, it is important to determine the marketable age of chickens for chilled sales. This study determined the effects of two marketable ages on the body measurements, carcass traits, and meat quality of yellow-feathered chickens. A total of 360 healthy one-day-old male Xueshan chickens were raised in six pens (straw-covered floor, numbered 1 to 6) and treated in the same manner (free access to food and water) until day 100. Sixty chickens from pens numbered 1 to 3 and 4 to 6 were selected to determine the body measurements, carcass traits, and meat quality at two slaughter ages (90 and 100 days), respectively. One hundred-day-old chickens had a higher body slope, cockscomb, keel, shank lengths, and higher live and dressed weights (*p* < 0.05). The abdomen skin follicle density, a*(redness) and b*(yellowness) values were higher in 100-day-old chickens (*p* < 0.05), whereas the 90-day-old chickens were characterized by better spotted skin. For the breast muscle, pH, shear force, a*, moisture, and protein and intramuscular fat contents were lower; moreover, L*(lightness) and b* were higher in 90-day-old chickens. In leg muscles, the pH, shear force, L*, b* and collagen content were lower; furthermore, the a* and moisture contents were higher in 90-day-old chickens (*p* < 0.05). These findings indicate that two marketable ages both have pros and cons, but 90 days chickens perform better on carcass appearance, and producers can adjust the marketable age to meet needs of different consumers. This study provides a unique idea and theoretical reference for breeding and marketing yellow-feathered chickens.

## 1. Introduction

China is a major chicken consumer globally, with an expected total chicken meat output of 15.1 million tons in 2022. Chicken consumption in China mainly consists of deep processed, white-feathered chickens and live poultry sales of yellow-feather broilers. In contrast with the European and American markets, where mostly white chickens are consumed, yellow-feathered chickens account for approximately 50% of chicken consumption in China. In recent years, industrial chain profits for yellow feathered chickens increased, whereas those for white feathered chickens decreased [1,2]. In addition, the yellow-feather chicken is an indigenous chicken breed in China, which is preferred by consumers. Moreover, the live poultry trade faces challenges due to the negative impact of COVID-19 and Avian Influenza, causing an increasing trend in processed chicken sales. In addition, consumers may prefer domestic to imported chickens and are willing to pay a premium for domestic chickens in some areas [3]. Several of the reasons mentioned above lead Chinese producers to increase the proportion of native, yellow-feathered chickens slaughtered and chilled for sale, thereby improving the production efficiency of chicken processing and helping meet consumer demand.

The production performance and meat quality of broilers are affected by many factors, including germplasm, diet, management, environment, and breeding age. Currently, the marketable ages of yellow-feathered chickens are flexible because age is a very easily adjustable factor that is closely related to the production performance of chickens. Many researchers have found that age has a strong effect on animal production. The slaughter age of chickens can affect carcass characteristics and meat quality [4,5]. Deng et al. [6] reported that the growth performance and different metabolites of yellow-feathered chickens significantly differed with the extension of the feeding period. Li et al. [7] suggested that the appropriate shortening of feeding time can improve myofiber density. Tok et al. [8] found that the feather quality of birds was related to age. Yuan et al. [9] and Li et al. [10] observed that age can influence vaccine susceptibility and thymus immunity in chickens. Park et al. [11] suggested that an optimized marketable age would benefit slaughter and chilling for the sales of broilers. Wang and Zhang [12,13] used metabolomics profiling to identify biomarkers in chilled chickens. Wang et al. [14] reported that chilled chickens were suitable for soft-boiled chickens, substituting for the hot–fresh carcass. The current research focuses on the effect of age on chicken production and the freshness of chilled chickens. However, few studies have examined the effect of marketable ages on carcass appearance in chilled for sale yellow-feathered chickens.

This study first investigated the difference between two marketable ages of yellow-feathered chickens on body measurement, slaughter performance, carcass appearance, and meat quality, ultimately identifying the ideal marketable age for slaughter and chilling for the sale of yellow-feathered chickens in current stage.

## 2. Materials and Methods

### 2.1. Ethics Statement

All experimental procedures were approved by the Institutional Animal Committee of Yangzhou University (approval number: YZUDWSY). All chickens were managed and handled according to the guidelines established and approved by Yangzhou University. All possible efforts were made to reduce the suffering of animals.

### 2.2. Animals and Experimental Design

A total of 2700 male one-day-old, yellow-feathered chickens (black beak, cyan–gray shank, and yellow feathers) were bred at Jiangsu Li-hua Animal Husbandry Stock Co., Ltd., China. All chickens were raised equally in six pens numbered 1 to 6 (500 × 1200 cm, 60 m^2^, and 7.5 birds/m^2^) with the same control during the experimental period. Each pen was equipped with 30 automatic drinking nipples and 15 feeders. The pens were provided with continuous lighting (24 h, 6 Lux, 2700 k), and the temperature was initially set at 29 °C for three days and reduced gradually by 1 °C per day until it reached 24 °C. The relative humidity was initially set at 70% and gradually reduced by 5% per week until it reached 55%. During the experimental period, all chickens had free access to food and water and were raised on the same diet (Table 1). All chickens were treated in exactly the same manner. On day 90, 180 healthy chickens were selected from three pens (60 per pen, numbered 1 to 3). On day 100, 180 healthy chickens were selected from three pens (60 per pen, numbered 4 to 6). The birds were anesthetized with sodium pentobarbital and slaughtered by manual exsanguination.

### 2.3. Body Measurements

At the marketing age (90 days), after 12 h of fasting, 60 males were randomly selected from each replicate, weighed (live weight, LW) with a portable electronic scale (Kaifeng, Jinghua, China), and then transferred to the adjacent slaughterhouse. On day 100, the same procedure was repeated. Cockscomb length (CCL) and cockscomb height (CCH) were measured in their natural state. Chest width (CW) was calculated as the distance between two shoulder joints. Chest depth (CD) was measured from the first thoracic vertebra to the front of the keel. Shank length (SL) was determined as the straight-line distance from the supra metatarsal joint to the middle of the third and fourth toes. The abovementioned measurements were performed using a slide caliper rule (Delixi Electric, Wenzhou, China), and the following measurements were performed using a soft tape measure (Deli, Ningbo, China). The body slope length (BSL) was gauged from the shoulder to the ipsilateral ischial tuberosity. Keel length (KL) was the distance, between the front and rear ends of the keel. Shank girth (SG) was the circumference of the middle of the shin. All operations were carried out while trying to keep the chicken calm, following agricultural standard NY/T823-2020 [15].

### 2.4. Slaughter Performance

Once all experimental chickens were measured, they were sent to the slaughterhouse. Chickens were electrically stunned in a water tank (240 mA, 120 V, 5 s) and then slaughtered via neck incision. After slaughter, the carcasses were cooled in a chilling room (4 °C). The de-feathered carcass, including the head and feet, was used to determine the dressed weight (DW). Chickens were eviscerated manually and weighted (semi-eviscerated weight: SEW) after the removal of the trachea, esophagus, gastrointestinal tract, crop, spleen, pancreas, gallbladder, and gonads. The head, feet, heart, liver, gizzard, glandular stomach, and abdominal fat were then removed and considered as the eviscerated weight (EW). Carcass yield was the percentage of live weight. The breast muscle, leg muscle, gizzard, testes, head, paws, and abdominal fat, including the leaf fat surrounding the cloaca and gizzard, were removed and weighed, and their weights are represented as BMW, TMW, GW, TW, HW, PW, and AFW, respectively. The breast muscle, leg muscle, head, and paw yields were computed as percentages of EW. The percentages of lean meat, gizzard, abdominal fat, and testes are represented as ((BMW + TMW)/EW) × 100, (GW/(GW + EW)) × 100, (AFW/(AFW + EW)) × 100, and (TW/(TW + EW)) × 100, respectively, following agricultural standard NY/T823-2020 [15].

### 2.5. Carcass Appearance

The follicle density, skin color, and spotted skin proportions of chickens were measured to evaluate carcass appearance, which was obtained in the de-feathered states. An orifice plate (2 × 2 cm^2^) was used to calculate the follicle density of the back midline and abdominal skin following a method similar to that described by Ji et al. [16]. Skin color in terms of lightness (L*), redness (a*), and yellowness (b*), which are the standards color measurements prescribed by the Commission Internationale de l’Éclairage, were measured via a chroma meter (CR-400, Konica Minolta, Tokyo, Japan) on the back skin. The data were calibrated using the tile (L* = 99.41, a* = −0.07, b* = −0.13). Both follicle density and skin color were measured three times and calculated as averages. The spotted skin proportions of chickens were divided into four grades (S, A, B, and C) according to quality, from high to low, based on the skin condition on the back and sides (Figure 1).

### 2.6. Meat Quality

Bilateral breast and leg muscle samples were collected to evaluate the meat quality at the marketing age. The left breast and leg muscles were measured to evaluate pH, shear force, water loss rate (WLR), and meat color. Right muscle samples were used to measure the proximate composition, including the moisture, protein, intramuscular fat (IMF), and collagen contents. The carcass appearance test was repeated three times, and the calculated average was the final reported value (n = 60).

The pH value was recorded at 1 h (pH_1_, 1 h after the muscle was collected) and 24 h (pH_24_, muscle was stored for 24 h at 4 °C) using a portable pH meter (pH-STAR, Matthaus, Berlin, Germany). The meat sample was cut to a depth of 1 cm. The pH meter electrode tip was inserted into the meat sample for measurement, and the calibration buffer (pH = 6.86) was used for processing and calibration before each measurement. Based on the procedures described by An et al. [17], water loss rates were measured by a meat quality pressure meter (Meat-1, Tenovo Food, Beijing, China) and shear force were measured by a digital tenderness meter (C-LM3B, Tenovo Food), respectively. To determine the shear force, the external fat and connective tissues were removed from samples (length = 6 cm, width = 3 cm, and height = 3 cm), which were then separately packed in plastic bags and cooked in a water bath at 80 °C (Memmert, Schwabach, Germany) until the core temperature reached 70 °C. After cooling to 4 °C (Refrigerator, Haier, Qingdao, China), samples were tested perpendicularly to the muscle fibers three times. To determine the water loss rate, 0.125 cm^3^ meat samples were wrapped with absorbent paper, and the pressure was set at 300 N, for 5 min [18]. Meat color, including lightness (L*), redness (a*), and yellowness (b*) as specified by the standard color measurements of the Commission Internationale de l´Éclairage, was measured using a chroma meter (CR-400, Konica Minolta, Tokyo, Japan). The data were calibrated using the tile (L* = 99.41, a* = −0.07, b* = −0.13). The proximate composition, including moisture, protein, IMF, and collagen, were measured after stripping exterior fat and connective tissue. Each sample was coarsely ground using a grinder to obtain a sample weighing approximately 200 g, which was analyzed using a near-infrared spectrophotometer according to Anderson et al. [19] (FOSS FoodScan 78800; Hilleroed, Denmark).

### 2.7. Statistical Analysis

Statistical analyses were performed using SPSS for Windows (version 22.0; SPSS Inc., Chicago, IL, USA). Data were analyzed using one-way analysis of variance (ANOVA). A *t*-test was used to analyze significant differences. *p* < 0.05 was set as the criterion for statistical significance.

## 3. Results and Discussion

### 3.1. Body Measurements

The effect of the marketable age on body measurements are presented in Table 2. Several measurements, including the cockscomb length, the body slope length, the keel length, and the shank length of broilers on day 100 were higher than those on day 90 (*p* < 0.05), indicating that an appropriate extension of rearing age can significantly improve the male body size and sexual characteristics and that broilers on day 100 are more economical for the live poultry trade. However, the results revealed no differences in the cockscomb height, the chest width, the chest depth, or the shank girth (*p* > 0.05). The results of body measurements implied that some body parts had reached their bottleneck on day 90. However, some still had growth potential, which may be a reference for producers of yellow-feathered broilers to determine a suitable marketable age. In fact, large-sized broilers are preferred over small-sized ones because the longer growing process tends to yield tastier meat [20]. Thus, broilers aged 100 days seemed to be better than those aged 90 days in terms of body measurements. However, according to Kim et al. [21], as the size of the chicken increases, the stocking density will also increase, which may lead to some adverse effects on the meat quality of chickens, causing their rejection by consumers. Moreover, the carcass characteristics of broilers aged 100 days have some disadvantages, which are detrimental to the sales of slaughtered and chilled chicken. From this perspective, broilers aged 90 days are better for producers in terms of processing and sales. Therefore, according to Lorets et al. [22] and Yaprak et al. [23], body measurements are important indicators for evaluating the growth of animals, and they are related to the slaughter performance, which is consistent with our results.

### 3.2. Slaughter Performance

Slaughter performance is considered a critical indicator of the economic profit obtained from meat animals [24]. Thus, the differences between the two marketing ages about slaughter performance were investigated. The carcass traits include the live weight, the dressed weight, the carcass yield, the semi-eviscerated yield, the eviscerated yield, the breast muscle yield, the testes yield, the gizzard yield, the head yield, and the paws yield of chickens, and the carcass yield, the semi-eviscerated yield, and the eviscerated yield were more than 90, 81, and 67%, respectively. The traits of slaughter performance presented in Table 3 show that the marketing age did not significantly affect the slaughter performance (*p* > 0.05), except for the live weight and the dressed weight of chickens (*p* < 0.05), which may be related to the closeness of the two slaughter days. The results indicate a lower slaughter performance for chickens on day 90 regarding some slaughter factors, despite a slightly lower fat content compared to chickens slaughtered on day 100. In addition, the carcass yield and semi-eviscerated yield of broilers on day 100 were slightly lower than those on day 90. The breast muscle, leg muscle, and lean meat yields of 100-day-old broilers were slightly higher than those on day 90, indicating that extending the slaughter age may be beneficial for lean meat production without significantly damaging the slaughter performance, in contrast with the results of Albuquerque et al. [25] and Seker et al. [26]. The differences between their results and ours may be related to the age and breed of chickens. Coban et al. [27] found that with an increased slaughter age, the neck yield (%) decreased and the leg yield (%) increased significantly in male Ross chickens. Mosca et al. [28] suggested that changing diets with different protein concentrations can improve the slaughter performance of chickens. In addition, it is worth mentioning that the testis yield and CCL on day 100 were higher than those on day 90, indicating that the sexual development of male, yellow-feathered broilers peaked between 90 and 100 days. Chickens on day 100 had a better lean meat production and a higher utilization rate than those on day 90, which is more suitable for frozen chickens. However, Hocking et al. [29] found that birds spent less time resting, more time foraging (pecking and scratching), and pecked more at the feather bunch at older ages. Sokolowicz et al. [30] found that age had a concomitant influence on the agonistic behavior of birds. According to the results of Coban et al. [27], profitability can be reduced as a result of decreased body weight gain and increased feed conversion rates. Xueshan chickens at the age of 100 days seemed to be more suitable for frozen products than those on day 90. However, we observed an increase in aggressive behavior among chickens. This is probably one of the reasons for the severely spotted skin leading to a low yield rate of high-quality frozen chickens, which is consistent with our results for carcass characteristics.

### 3.3. Carcass Appearance

Carcass characteristics of chickens do not affect live poultry trading. However, they significantly impact industrially processed products. Wu et al. [31] found that carcass characteristics of chickens were the most intuitive sensory indicators influencing the choice of consumers. The skin color, uniformity, and color consistency of poultry were important reference attributes for consumers to choose and buy. Currently, research on the carcass characteristics of yellow-feathered broilers is mainly limited to skin hair follicles [16,32]. Therefore, three parameters related to the slaughter characteristics of yellow-feathered broilers were examined, including the follicle density, the skin color, and the spotted skin proportions. As shown in Table 4, the abdominal follicle density of chickens on day 100 was higher than that on day 90 (*p* < 0.05), but the difference was not significant on the back skin (*p* > 0.05). The lower the follicle density, the smoother the skin, and the more favored by consumers, indicating that 90-day-old chickens had a better performance in terms of follicle density than 100-day-old broilers. In addition, Zhang et al. [33] reported that feather follicles on chicken skin could promote the bacterial cross-contamination of carcasses, indicating that 100-day-old chickens may be more susceptible to pollution during slaughter than 90-day-old chickens. For skin color, the a^*^ value of broilers was higher at day 90 (*p* < 0.05), while L* and b* values did not significantly differ (*p* > 0.05), indicating that the skin of 90-day-old broilers were more rosy relative to chickens aged 100 days, which may be attributed to their genotype [34]. Visual observation showed that chickens on day 90 were brighter, and those on day 100 were more yellow and red. Therefore, the 100-day-old chickens more closely resemble the traditional Chinese chicken, which may influence consumer choice, thereby improving sales performance. Furthermore, regarding spotted skin proportions, chickens on day 90 had higher S and C ratios and lower A and B ratios. These results show that chickens at the age of 90 days performed better In terms of spotted skin proportion than 100-day-old broilers.

In general, chickens on day 90 performed better than those on day 100 in terms of carcass characteristics, which can significantly affect the choice of consumers [31]. However, chickens on day 100 performed better in terms of slaughter performance with better carcass characteristics, as in the follicle density and spotted skin. Chickens at the age of 90 may be more suitable for chilled sales.

### 3.4. Meat Quality

Meat quality is the most direct indicator of poultry meat quality. It is generally reflected by several traits such as pH, water loss rate, meat color, shear force, and proximate composition. The effects of marketing age on meat quality traits are shown in Table 5. In terms of physical traits, the pH_1_ and pH_24_ values of chickens on day 100 were higher than those on day 90 (*p* < 0.05). pH is an important reference for evaluating muscle quality and is an important indicator of the speed of muscle glycogenolysis after poultry slaughter, which is mainly affected by phosphofructokinase activity [35]. The amount of lactic acid produced in the muscle is related to the glycogen content in the muscle. The more glycogen reserves in the muscle, the more lactic acid is produced during intense exercise, the more accumulated, and the greater the drop in pH, which leads to the acidification of meat. In addition, a previous study showed that the pH of broiler muscle was inversely correlated with lightness, yellowness, and shear force [36]. Huo et al. [37] reported differences in muscle fiber composition, density, and diameter, which can affect muscle pH. In our study, chickens on day 90 had a lower pH, suggesting that the birds have different muscle fiber compositions and may be subjected to greater slaughter stress, resulting in altered rates of glycolysis and more lactic acid accumulation compared with those on day 100. In general, the pH value of meat is directly related not only to muscle acidity but also to meat color, fiber composition, and shear force [38]. The shear force values of broilers on day 100 were higher than those on day 90 (*p* < 0.05), consistent with the results of Li et al. [7], this may be related to the growth of muscle fibers [39]. Meat color is an intuitive meat quality parameter. The breast muscle of chickens on day 100 had a higher a* value and lower L* and b* values than those on day 90 (*p* < 0.05), but this was reversed in the leg muscle (*p* < 0.05), indicating that chickens on day 100 have a darker pectoral breast muscle and a brighter leg muscle, which may be due to differences in pigmentation, myoglobin, and hemoglobin contents. According to Garmiene et al. [40], the sensory (visual) evaluation of chickens on day 100 scored higher, which may influence the choice of Chinese consumers. In addition, the water loss rate is generally used to represent the water-holding capacity. A lower water-holding capacity in muscles can lead to the loss of nutrients and flavor, resulting in a decline in meat quality. Li et al. [7] found that the muscle shear force increases with age. There was no significant difference between the water loss rate of chickens on days 90 and 100 (*p* > 0.05), which may be due to the two congenial slaughter ages.

In terms of the proximate composition, moisture, protein, IMF, and collagen contents of the breast muscle were higher on day 100. Moisture content in the leg muscle was lower compared to those on day 90 (*p* < 0.05). A higher IMF content reflects the succulence, flavor, and nutritional value of meat [41]. In the study, most metrics were higher on day 100 than on day 90. The breast muscle IMF content of chickens on day 100 was more than twice that of chickens at the age of 90, indicating a higher nutrient content in the muscle of 100-day-old chickens. Qiu et al. [42] suggested that IMF is positively correlated with tenderness, juiciness, and flavor, which are related to the prevailing for traditional native chickens in China. Thus, a higher IMF was used as a reference in marketing age determination. In addition, the moisture content of leg muscles of chickens on day 100 was lower than that on day 90 (*p* < 0.05), which may be due to the increased activity of chickens on day 100 based on the results of cockscomb length and testes yield. Meat from chickens on day 100 may require hard chewing by consumers due to its higher shear force in muscles and lower moisture content in leg muscle than on day 90 (*p* < 0.05). Furthermore, the protein content in breast muscle of the chickens on day 100 was higher than that on day 90 (*p* < 0.05).

## 4. Conclusions

In conclusion, our results support extending the marketable age to 100 days to improve slaughter performance in terms of eviscerated, breast muscle, and leg muscle yields, there are also benefits to meat quality with respect to protein and IMF contents. However, shortening the marketable age to 90 days enhances the quality of carcass characteristics such as follicle density, skin color, and spotted skin proportions; moreover, it can improve physical meat quality traits such as the pH and shear force. The present study shows the advantages and disadvantages of two marketable ages for slaughtering yellow-feathered broilers by determining body measurements, slaughter performance, carcass appearance, and meat quality. The results suggest that adjusting the marketable age can be an effective approach to drive chilled sales in indigenous, yellow-feathered chickens and the producers of yellow-feathered chickens can attract consumers from different regions through the different marketable ages. In addition, yellow-feathered Xueshan chickens on day 90 have a better performance on spotted skin, which is one of the most intuitive subjective references for consumers, based on our current study. However, the aspects of the breeding cost, breeding benefit, nutritional value, and flavor of muscle should be supplemented with further analyses. Our future studies will examine these aspects to uncover novel strategies to obtain high-quality products, yielding increased profits for producers and additional benefits for consumers.

## Figures and Tables

**Figure 1 animals-12-02422-f001:**
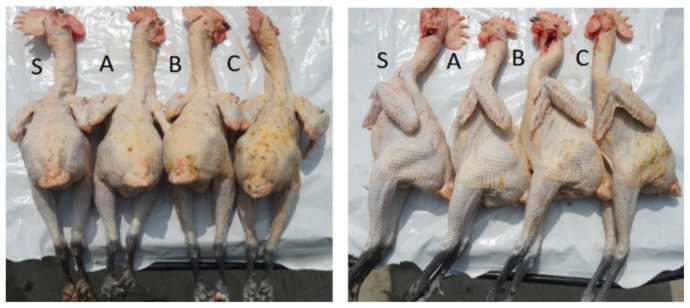
Conditions of spotted skins on the back and sides of chickens on day 90. Carcasses were divided into four grades (S, A, B, and C) according to the proportion of spotted skin: S: extremely sparsely spotted skin, A: slightly spotted skin, B: medium spotted skin, and C: conspicuously spotted skin.

**Table 1 animals-12-02422-t001:** Compositions and nutritional contents of the experimental diets.

Items	1–28 d	29–63 d	64–100 d
Ingredient (%)			
Corn	55.82	62.49	69.30
Flour	2.00	2.00	2.00
Soybean meal	33.10	25.20	17.20
Corn protein flour	1.00	2.00	3.00
Soybean oil	1.31	1.96	2.27
Stone powder	1.41	1.24	1.25
Calcium hydrogen phosphate	1.36	1.11	0.98
Choline chloride	1.00	1.00	1.00
Premix	3.00	3.00	3.00
Nutritional level (%)			
Crude protein	21.00	18.50	16.00
Metabolizable energy (MJ/kg)	12.13	12.55	12.97
Ca	0.94	0.80	0.75
Available phosphorus	0.38	0.33	0.30
Digestible lysine	1.05	0.90	0.80

Premix provided for each kilogram of diet: V_A_: 7500 IU, V_D_: 3000 IU, V_E_: 50 IU, V_K3_: 50 mg; V_B1_: 90 mg; V_B2_: 300 mg; V_B6_: 60 mg; V_B12_: 0.4 mg; V_B3_: 1000 mg; V_B5_: 300 mg, folate: 20 mg; biotin: 2.0 mg; Fe: 1.3 g; Cu: 0.25 g; Zn: 2.0 g; Mn: 2.35 g; I: 20.0 mg; and Se: 4.5 mg. Except for crude protein, whose value was measured, the levels of all nutrients were calculated 1.

**Table 2 animals-12-02422-t002:** Effects of marketable ages on body measurements of broilers on days 90 and 100.

	Marketing Age			
Items	90 d	100 d	SEM	*p*-Value
Cockscomb length (cm)	7.69 ^b^	8.92 ^a^	0.174	<0.001
Cockscomb height (cm)	4.11	4.27	0.063	0.079
Body slope length (cm)	22.36 ^b^	23.22 ^a^	0.125	<0.001
Keel length (cm)	14.86 ^b^	16.69 ^a^	0.159	<0.001
Chest width (cm)	8.20	8.36	0.057	0.208
Chest depth (cm)	7.29	7.66	0.079	0.390
Shank length (cm)	8.83 ^b^	9.64 ^a^	0.081	<0.001
Shank girth (cm)	4.35	4.38	0.020	0.334

^a, b^ Within a row, for each factor, different superscripts indicate significant differences (*p* < 0.05). Data represent three replicates, with 60 birds per replicate. SEM: Standard error of the mean.

**Table 3 animals-12-02422-t003:** Effects of marketable ages on slaughter performance of broilers on days 90 and 100.

	Marketing Ages			
Items	90 d	100 d	SEM	*p*-Value
Carcass yield (%)	91.21	90.72	0.182	0.185
Semi-eviscerated yield (%)	81.72	81.00	0.333	0.279
Eviscerated yield (%)	67.72	67.80	0.303	0.894
Breast muscle (%)	18.01	18.84	0.382	0.284
Leg muscle (%)	23.56	24.84	0.475	0.179
Lean meat (%)	41.57	43.68	0.802	0.192
Gizzard (%)	2.65	2.35	0.086	0.079
Abdominal fat (%)	2.90	2.93	0.208	0.931
Testis (%)	1.60	1.86	0.121	0.291
Head (%)	6.39	5.87	0.149	0.085
Paws (%)	0.53	0.55	0.026	0.672
Live weight (g)	1942.54 ^b^	2254.52 ^a^	24.152	<0.001
Dressed weight (g)	1771.93 ^b^	2045.22 ^a^	21.613	<0.001

^a, b^ Within a row, for each factor, different superscripts indicate significant differences (*p* < 0.05). Data represent three replicates, with 60 birds per replicate. Carcass yield, % = carcass weight/live weight × 100; eviscerated yield, % = eviscerated weight/live weight × 100; semi-eviscerated yield, % = semi-eviscerated weight/live weight × 100; breast muscle yield, % = breast muscle weight/eviscerated weight × 100; leg muscle yield, % = leg muscle weight/eviscerated weight × 100; lean meat yield, % = (breast muscle weight + leg muscle weight)/eviscerated weight × 100; gizzard yield, % = gizzard weight/(gizzard weight + eviscerated weight) × 100; abdominal fat yield, % = abdominal fat weight/(abdominal fat weight + eviscerated weight) × 100; head yield, % = head weight/eviscerated weight × 100; paw yield, and % = paw weight/eviscerated weight × 100. SEM: Standard error of the mean.

**Table 4 animals-12-02422-t004:** Effects of marketable ages on carcass characteristics of broilers on days 90 and 100.

		Marketing Ages			
Items		90 d	100 d	SEM	*p*-Value
Follicle density (piece/cm^2^)	Back	4.33	4.54	0.080	0.177
	Abdomen	3.41 ^b^	3.97 ^a^	0.070	<0.001
Skin color	L*	72.37	72.62	0.276	0.656
	a*	16.44 ^a^	11.74 ^b^	0.480	<0.001
	b*	19.76 ^a^	15.87 ^b^	0.711	0.005
Spotted skin level proportions (%)	S	37.93	13.33		
	A	17.24	33.33		
	B	24.14	33.33		
	C	20.69	20.00		

^a, b^ Within a row, for each factor, different superscripts indicate significant differences (*p* < 0.05). Data represent three replicates, with 60 birds per replicate (n = 180). L*: lightness; a*: redness; b*: yellowness. S: good, extremely sparsely spotted skin; A: fair, slightly spotted skin; B: poor, medium spotted skin; C: very poor, conspicuously spotted skin. SEM: standard error of the mean.

**Table 5 animals-12-02422-t005:** Effects of marketable ages on meat quality of broilers between day 90 and 100.

Items	Marketing Ages	pH		Shear Force (N)	Water Loss Rate (%)	Meat Color			Proximate Composition			
		pH_1_	pH_24_			L*	a*	b*	Moisture (%)	Protein (%)	Intramuscular Fat (%)	Collagen (%)
Breast muscle	90 d	5.92 ^b^	5.95 ^b^	15.98 ^b^	25.39	52.93 ^a^	2.64 ^b^	15.03 ^a^	72.43 ^b^	24.77 ^b^	0.51 ^b^	0.52
	100 d	6.20 ^a^	6.31 ^a^	23.09 ^a^	24.69	45.78 ^b^	15.09 ^a^	12.94 ^b^	72.82 ^a^	25.50 ^a^	1.22 ^a^	0.52
	SEM	0.031	0.039	1.268	1.219	1.035	1.322	0.504	0.092	0.090	0.076	0.025
	*p*-value	<0.001	<0.001	0.006	0.782	<0.001	<0.001	0.036	0.031	<0.001	<0.001	0.928
Leg muscle	90 d	6.22 ^b^	6.31 ^b^	15.49 ^b^	20.02	44.18 ^b^	16.68 ^a^	12.41 ^b^	74.58 ^a^	21.59	2.80	0.45 ^b^
	100 d	6.58 ^a^	6.60 ^a^	25.88 ^a^	21.49	56.67 ^a^	1.51 ^b^	15.57 ^a^	73.83 ^b^	21.64	2.98	0.61 ^a^
	SEM	0.047	0.034	1.534	0.892	1.329	1.446	0.471	1.221	0.081	0.078	0.035
	*p*-value	<0.001	<0.001	0.001	0.417	<0.001	<0.001	<0.001	0.001	0.801	0.251	0.018

^a, b^ Within a column for each factor, different superscripts indicate significant differences (*p* < 0.05). Data represent six replicates, with 60 birds per replicate (n = 180). L*: lightness; a*: redness; b*: yellowness. pH_1_: pH value measured 1 h after slaughter. pH_24_: pH value measured 24 h after slaughter. Water loss rate, % = (W _Initial_ − W _Final_)/W _initial_ × 100. SEM: standard error of the mean.

## Data Availability

All available data are incorporated in the manuscript.

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
