# Peer review of "Indigenous, Yellow-Feathered Chickens Body Measurements, Carcass Traits, and Meat Quality Depending on Marketable Age"

_animals, 2022, doi:10.3390/ani12182422_

Round 1

Reviewer 1 Report

The manuscript addressed the effects of two marketable ages on body measurements, slaughter performance, and meat quality in indigenous yellow-feathered broilers. The work has merit to Animals readers and industry in general; however, the manuscript has weakness that needs to be addressed.

To sum up, The detection indicators in this study are too simple to fully support the conclusions drawn by the authors. From the results obtained in the article, it is not clear that the meat quality was better on day 90 and more suitable for slaughter and chilled sales. It is suggested to make further evaluation from the aspects of breeding cost, breeding benefit, nutritional value and flavor of muscle

The authors should consult with someone skilled in English language scientific writing to address language issues.

Title: Title is too complicated, Please reconsider and modify it.

The Abstract disseminates scientific information through abstracting journals and through convenience for readers without reference to the body of the manuscript. This reviewer finds the abstract lacking enough details for the experimental procedures used (duration of the experiment, feeding program, sampling procedure, etc); this information needs to be added in the revised manuscript.

Introduction:

It is suggested to supplement the research progress related to this study and what is the innovation of this study?

M&M

Statistical Analysis: “Data were analyzed using one-way analysis of variance (ANOVA)”, Would you like to ensure using one-way analysis of variance is correct in here?

Results and discussion

Good.

Conclusion: Conclusions need to be more precise. I don't think this conclusion can be fully drawn from this study.

Author Response

  1. To sum up, The detection indicators in this study are too simple to fully support the conclusions drawn by the authors. From the results obtained in the article, it is not clear that the meat quality was better on day 90 and more suitable for slaughter and chilled sales. It is suggested to make further evaluation from the aspects of breeding cost, breeding benefit, nutritional value and flavor of muscle.

Response 1: Thank you for your comments and suggestions. In production experiments, multiple factors are very complex and easy to influence each other. Based on the current data, we preliminarily judged that 90-day-old chickens have better skin and are more suitable for slaughter and chilled sales. Because carcass appearance is the most direct observation by consumers. The fatty acids, amino acids, ATP, IMP, etc. in muscles and the aspects of breeding cost will be considered as suggested in the future study.

  1. The authors should consult with someone skilled in English language scientific writing to address language issues.

Response 2: Thank you for your comments. We have addressed the language issues.

  1. Title: Title is too complicated, Please reconsider and modify it.

Response 3: Thank you for your comments. We have revised the title to make it shorter and simple.

  1. The Abstract disseminates scientific information through abstracting journals and through convenience for readers without reference to the body of the manuscript. This reviewer finds the abstract lacking enough details for the experimental procedures used (duration of the experiment, feeding program, sampling procedure, etc); this information needs to be added in the revised manuscript.

Response 4: Thank you for your comments. We have rewritten the abstract to make it more precise.

  1. M&M

Statistical Analysis: “Data were analyzed using one-way analysis of variance (ANOVA)”, Would you like to ensure using one-way analysis of variance is correct in here?

Response 5: Thank you for your comments. The tests in this article only used one factor (age of chickens). However, follicle density in back and abdomen have not been compared together, and breast and leg muscles are also not put together for comparison. Therefore, General Linear Model and ANOVA are all applicable.

  1. Conclusion: Conclusions need to be more precise. I don't think this conclusion can be fully drawn from this study.

Response 6: Thank you for your comments. We have revised the conclusion as suggested. Please see line 372-390.

Reviewer 2 Report

The purpose of this study was to determine the effects of different marketable ages of yellow- feathered broilers on body measurement, slaughter performance, carcass appearance, and  meat quality, ultimately identifying the ideal marketable age for slaughter and chilled sales of yellow-feathered broilers. The Introduction chapter provides an overview of the world's knowledge on this subject. The material used in the research is sufficiently numerous, but some supplementing the description in Materials and Methods chapter are needed. The results are described usually correctly. The discussion is exhaustive. Summary of the results are correct. Some corrections are needed before publishing an article in Animals. The proposed changes are listed below.

General comments:

Please prepare the article in accordance with the instructions for authors:

On the 1st page on the left side, add "Citaion" and the required data

For significance please use lowercase "p" in italic instead of "p" throughout the main article in some places

The references chapter for page ranges use long (-) from the symbol function, instead of short (-) from the keyboard

After the names of journal without abbreviaotion form delete "a dot." before the volume number

Detailed comments:

L20 water loss rate = drip loss? Do higher meat water loss mean better meat quality?

L29 „L*, and a* values ​​of braest and thigh muscles were higher at day 100”? - no compliance with the data in the tables

L30 moisture thigh meat also?

L32 „whereas b* values ​​were higher” - no compliance with the data in the table

L90 Was that for humidity?

L90 add information about the photoperiod (length, color, intensity light)

In table 1 The sum of ingredients must be 100% in diets used 1-28d, 29-63d, 64-100d, it is 100.9%, please correct it to a corn percentage.

„Metabolizable” instead of Metabolic

L104 scales name, manufacturer data use to determine BW

L105 as measured for CCH and CCL

L119 will the intensity and voltage parmeters be too big?

L120 How long were the carcasses chilled, is the measurement 1h from slaughter possible?

L126 thigh? or leg muscle? (thigh plus drumstick muscles)

L129 leg instead of thigh

L154 WLR or drip loss (DL)?

L160 „An et al. [14]” instead of current form

L160 provide the pH of the calibration buffers, the method of measuring the pH

L171 enter the tile's calibration data (Y =, x =, y =), the diameter of the measurement field

L174 200 g thigh meat or 200 g leg meat

L188 full names as in Table 2 instead of CCL, BSL, KL, SL

L192 Complete names instead of CCH, CW, CD, SG

In tables 2-4 90 d, 100 d or d 90, d 100 in header and Itams, „p-Value” for tables 2-4, p in italic, Value from uppercase

In table 2 - 0.390, three decimal places for p-Value

L208 p in italc

L 214 full name for LW, DW, CW, SEW, EW, BMW, TMW, GW, TW, HW, PW

L217 full name for LW and DW

L 222 leg muscle?

IN table 3 without "yield" from "Breast muscle (%)" to Paws (%) "three decimal places for p-Value <0.001 instead of <0.0001; The first feature is Live weight (g), the second is Dressing weight (g), the third is "Carcass yield" ... ..

L245 p in italic

L257 Wu et al. [28] instead of current form

L 278 „…S and C ratios and lower A and B ratios” - instaed of current form

L296 DL or WLR?

L302 why pH24 higher than pH1

L303 shear stress?, or slaughter stress

L307 and what about influencing what? See Table 5

L320 Li et al. [7] instead of current form

L337 describe the influence of physical activity on glycogen concentration - production of lactic acid and acidification of meat

L323-325 description inconsistent with the results L327 of chickens or Brest meat?

L335 lower moisture, breast meat also? compared to what?

L337 why higher meat CP levels in older males?

L378 no chapter: Informed Consent Statement

L384 Provide abbreviation name journal

L386 Agribussiness 2020, instead of the curent form

L394 "Animals 2020, 10, article number" instead of the current form

L405 Ji, GG .; Zhang, M ,; Liu, Y.;… ..

L409 Journal of J. …… - correct name is required

L414 „J. AOAC Int.” is the correct abbreviation?

L L438, 440. 454, 456, 458, 460, 467 article or page number are required

L443 without a one „dot” before Br.

L460 „Braz. J. Poult. Sci.” instaed of cureent form

Author Response

  1. On the 1st page on the left side, add "Citaion" and the required data

Response 1: Thank you for your comments. We have added "Citaion" and the required data.

  1. For significance please use lowercase "p" in italic instead of "p" throughout the main article in some places

Response 2: Thank you for your comments. We have revised the errors in the manuscript.

  1. The references chapter for page ranges use long (-) from the symbol function, instead of short (-) from the keyboard

Response 3: Thank you for your comments. We have revised the errors in the manuscript.

  1. After the names of journal without abbreviaotion form delete "a dot." before the volume number

Response 4: Thank you for your comments. We have revised the errors in the manuscript.

  1. L20 water loss rate = drip loss? Do higher meat water loss mean better meat quality?

Response 5: Thank you for your comments. Water loss rate does not equal drip loss. We have revised the errors in the manuscript. We used water loss rate to describe the ability of meat to retain moisture, it is detected by external force extrusion, while drip loss does not exert any external force, only under the action of gravity, the liquid loss of the protein system. The lower the water loss rate, the easier it is for the meat to retain its original flavor during storage and transportation.

  1. L29 „L*, and a* values of breast and thigh muscles were higher at day 100”? - no compliance with the data in the tables.

Response 6: Thank you for your comments. We have revised the errors in the abstract. Please see line 37-40.

  1. L30 moisture thigh meat also?

Response 7: Thank you for your comments. We have revised the errors in the abstract. Please see line 37-40.

  1. L32 „whereas b* valueswere higher” - no compliance with the data in the table

Response 8: Thank you for your comments. We have revised the errors in the abstract. Please see line 37-38.

  1. L90 Was that for humidity?

Response 9: Thank you for your comments. The proper humidity is very important for chicken growth. Our experimental phase is from March to June in Changzhou, Jiangsu, eastern China.

  1. L90 add information about the photoperiod (length, color, intensity light)

Response 10: Thank you for your comments. We have added information about the photoperiod. Please see Animals and Experimental Design. Please see line 96-97.

  1. In table 1 The sum of ingredients must be 100% in diets used 1-28d, 29-63d, 64-100d, it is 100.9%, please correct it to a corn percentage.

Response 11: Thank you for your comments. We have revised it in the manuscript. Please see table 1 ( line 106-107).

  1. „Metabolizable” instead of Metabolic

Response 12: Thank you for your comments. We have revised it in the manuscript. Please see table 1 ( line 106-107).

  1. L104 scales name, manufacturer data use to determine BW

Response 13: Thank you for your comments. We have added the information in the manuscript. Please see line 113.

  1. L105 as measured for CCH and CCL

Response 14: Thank you for your comments. We have revised it in the manuscript. Please see line 115.

  1. L119 will the intensity and voltage parmeters be too big?

Response 15: Thank you for your comments. The standard voltage in China is 220 V and 120 V in America. This can reduce stress when the chicken enters the water. Referring to the method of Huo et al.

Huo, W.; Weng, K.; Gu, T.; Zhang, Y.; Zhang, Y.; Chen, G.; Xu, Q. Effect of muscle fiber characteristics on meat quality in fast- and slow-growing ducks. Poult. Sci. 2021, 100, 101264.

  1. L120 How long were the carcasses chilled, is the measurement 1h from slaughter possible?

Response 16: Thank you for your comments. We put the carcass in the chilled room for 10 minutes, and control the measurement of slaughter performance indicators within 1 hour. We had 50 people in attendance, including people from our school and the company.

  1. L126 thigh? or leg muscle? (thigh plus drumstick muscles)

Response 17: Thank you for your comments. It is leg muscle. We have revised it in the manuscript.

  1. L129 leg instead of thigh

Response 17: Thank you for your comments. We have revised it in the manuscript. Please see line 136-137.

  1. L154 WLR or drip loss (DL)?

Response 17: Thank you for your comments. It is water loss rate. 

  1. L160 „An et al. [14]” instead of current form

Response 20: Thank you for your comments. We have revised it in the manuscript. Please see line 174.

  1. L160 provide the pH of the calibration buffers, the method of measuring the pH

Response 21: Thank you for your comments. The pH of the calibration buffer is 6.86. We have added the information in the manuscript. Please see line 172-173.

  1. L171 enter the tile's calibration data (Y =, x =, y =), the diameter of the measurement field

Response 22: Thank you for your comments. Calibrate the data using the tile (L* = 99.41, a* = −0.07, b* = −0.13). We have added the information in the manuscript. Please see line 186-187.

  1. L174 200 g thigh meat or 200 g leg meat

Response 23: Thank you for your comments. It is 200 g leg meat.

  1. L188 full names as in Table 2 instead of CCL, BSL, KL, SL

Response 24: Thank you for your comments. We have revised it in the manuscript. Please see line 203-204.

  1. L192 Complete names instead of CCH, CW, CD, SG

Response 25: Thank you for your comments. We have revised it in the manuscript. Please see line 207-208.

  1. In tables 2-4 90 d, 100 d or d 90, d 100 in header and Itams, „p-Value” for tables 2-4, p in italic, Value from uppercase

Response 26: Thank you for your comments. We have revised it in the manuscript. Please see line 223-224, 262-263, 304-305.

  1. In table 2 - 0.390, three decimal places for p-Value

Response 27: Thank you for your comments. We have revised it in the manuscript. Please see line 223-224.

  1. L208 p in italc

Response 28: Thank you for your comments. We have revised it in the manuscript. Please see line 224.

  1. L 214 full name for LW, DW, CW, SEW, EW, BMW, TMW, GW, TW, HW, PW

Response 29: Thank you for your comments. We have revised it in the manuscript. Please see line 229-230.

  1. L217 full name for LW and DW

Response 30: Thank you for your comments. We have revised it in the manuscript. Please see line 234-235.

  1. L 222 leg muscle?

Response 31: Thank you for your comments. Yes, it is leg muscle, we have revised it in the manuscript. Please see line 29-240.

  1. IN table 3 without "yield" from "Breast muscle (%)" to Paws (%) "three decimal places for p-Value <0.001 instead of <0.0001; The first feature is Live weight (g), the second is Dressing weight (g), the third is "Carcass yield" ... ..

Response 32: Thank you for your comments. We have revised it in the manuscript. Please see line 262-263.

  1. L245 p in italic

Response 33: Thank you for your comments. We have revised it in the manuscript. Please see line 263.

  1. L257 Wu et al. [28] instead of current form

Response 34: Thank you for your comments. We have revised it in the manuscript. Please see line 275.

  1. L 278 „…S and C ratios and lower A and B ratios” - instaed of current form

Response 35: Thank you for your comments. We have revised it in the manuscript. Please see line 296-297.

  1. L296 DL or WLR?

Response 36: Thank you for your comments. It is water loss rate..

  1. L302 why pH24 higher than pH1

Response 37: Thank you for your comments. In normal animal muscle, glycolysis is slow after death, and lactic acid in muscle is gradually accumulated, resulting in a slow decline in pH in muscle. In our experiments, each meat sample was sealed in a small individual clear bag and stored at 4°C, which resulted in some oozing liquid (drip loss) when we measured pH at 24 hours, these fluids have a lower pH, which may carry away the lactic acid produced at 1 hour. Meat that may have gone bad has been culled. p-value was not significant between pH1 and pH24.

  1. 303 shear stress?, or slaughter stress

Response 38: Thank you for your comments. It is shear stress, we have changed it to shear force. Please see line 325.

  1. L307 and what about influencing what? See Table 5

Response 39: Thank you for your comments. We have revised the errors. Please see line 327-330.

  1. L320 Li et al. [7] instead of current form

Response 40: Thank you for your comments. We have revised it in the manuscript. Please see line 342.

  1. L337 describe the influence of physical activity on glycogen concentration - production of lactic acid and acidification of meat

Response 41: Thank you for your comments. We have added the information in the manuscript. Please see line 320-324.

During physical activity, the lactic acid produced by the muscles quickly enters the blood, and is first continuously oxidized to pyruvate in the liver, and then enters the citric acid cycle for complete oxidation, which releases a large amount of energy at the same time. The excess lactate that is not oxidized in the liver is converted into pyruvate by lactate dehydrogenase, and then generates glucose through the pyruvate branch. When there is a large amount of lactic acid, the liver does not have time to oxidize and convert it, resulting in the accumulation of lactic acid in the muscle, resulting in a decrease in muscle pH. The amount of lactic acid produced in the muscle is related to the glycogen content in the muscle. Muscle is the tissue that stores the most glycogen and has the best nutritional status. The more glycogen reserves in the muscle, the more lactic acid is produced during intense exercise, the more accumulated, and the greater the drop in pH,which leads to acidification of meat.

  1. L323-325 description inconsistent with the results L327 of chickens or Brest meat?

Response 42: Thank you for your comments. We have revised the errors. Please see line 346-348.

  1. L335 lower moisture, breast meat also? compared to what?

Response 43: Thank you for your comments. We have revised the errors. Please see line 357-360.

  1. L337 why higher meat CP levels in older males?

Response 44: Thank you for your comments. We have revised the errors. Please see line 357-361.

  1. L378 no chapter: Informed Consent Statement

Response 45: Thank you for your comments. We have added the information in the manuscript. Please see line 406.

  1. L384 Provide abbreviation name journal

Response 46: Thank you for your comments. We have added the information in the manuscript. Please see line 412.

  1. L386 Agribussiness 2020, instead of the curent form

Response 47: Thank you for your comments. We have revised the errors. Please see line 414-415.

  1. L394 "Animals 2020, 10, article number" instead of the current form

Response 48: Thank you for your comments. We have revised the errors. Please see line 422-423.

  1. L405 Ji, GG .; Zhang, M ,; Liu, Y.;… ..

Response 49: Thank you for your comments. We have revised the errors. Please see line 440-442.

  1. L409 Journal of J. …… - correct name is required

Response 50: Thank you for your comments. We have revised the errors. Please see line 443.

  1. L414 „J. AOAC Int.” is the correct abbreviation?

Response 51: Thank you for your comments. It should be correct. We search the abbreviation from the CASSI.

https://www.webofscience.com/wos/alldb/full-record/WOS:000248520200028

https://cassi.cas.org/search.jsp

  1. L L438, 440. 454, 456, 458, 460, 467 article or page number are required

Response 52: Thank you for your comments. We have added the information in the manuscript. Please see the references (line 410-501 ).

  1. L443 without a one „dot” before Br.

Response 53: Thank you for your comments. We have revised the errors. Please see line 480.

  1. L460 „Braz. J. Poult. Sci.” instaed of cureent form

Response 54: Thank you for your comments. We have revised the errors. Please see line 497.

Round 2

Reviewer 1 Report

Thank you for your reply!

Author Response

Dear reviewer, thank you for your review, I think these comments are very valuable, I have revised as you requested, thank you again for your comments.